# A Comparative Analysis of COVID-19 Response Measures and Their Impact on Mortality Rate

Tomokazu Konishi

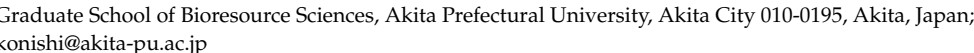

Graduate School of Bioresource Sciences, Akita Prefectural University, Akita City 010-0195, Akita, Japan; konishi@akita-pu.ac.jp

**Abstract:** (1) Background: The coronavirus disease 2019 (COVID-19) pandemic significantly affected the population worldwide, with varying responses implemented to control its spread. This study aimed to compare the epidemic data compiled by the World Health Organization (WHO) to understand the impact of the measures adopted by each country on the mortality rate. (2) Methods: The increase or decrease in the number of confirmed cases was understood in logarithmic terms, for which logarithmic growth rates "*K*" were used. The mortality rate was calculated as the percentage of deaths from the confirmed cases, which was also used for logarithmic comparison. (3) Results: Countries that effectively detected and isolated patients had a mortality rate 10 times lower than those who did not. Although strict lockdowns were once effective, they could not be implemented on an ongoing basis. After their cancellation, large outbreaks occurred because of medical breakdowns. The virus variants mutated with increased infectivity, which impeded the measures that were once effective, including vaccinations. Although the designs of mRNA vaccines were renewed, they could not keep up with the virus mutation rate. The only effective defence lies in steadily identifying and isolating patients. (4) Conclusions: these findings have crucial implications for the complete containment of the pandemic and future pandemic preparedness.

**Keywords:** logarithmic growth rates; lockdown; vaccination; containment of the pandemic; identifying and isolating patients

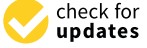



## 1. Introduction

The coronavirus disease 2019 (COVID-19) pandemic has spread worldwide, leading to physical disabilities and countless deaths [1–6]. Countries implemented various measures against it; measures taken globally by 2022 are reported here [7]. By 2023, many countries had ceased these measures, but until then, they had adopted a range of strategies, including societal enforcements such as lockdowns, movement restrictions, office closures, quarantines, and individual precautions like maintaining distance and wearing masks. The World Health Organization (WHO) and the European Centre for Disease Prevention and Control recommend vaccines first, followed by non-medical measures that can be taken locally and individually [8,9].

Studies examining the actual effects of such measures require accurate measurements and proper analyses [10]. In the case of people's movement, stringent measures such as lockdowns effectively restricted people's movement, whereas recommendations for restraint such as mild physical distancing were observed to reduce traffic [11]. In Japan, the government urged people to avoid nighttime outings and travel. This led to a considerable reduction in people's movements; however, the number of patients and deaths did not significantly change [12]. Although there is a mild request for self-restraint, maintaining distance can be challenging for vulnerable population groups [13].

There were significant differences in isolation measures from country to country [14], and their effectiveness varied. Lockdowns and shelter-in-place orders in Europe and the United States have done little to reduce mortality rates, although they involve strong

enforcement [15]. Some countries, such as Iceland and Taiwan, successfully contained the virus by conducting frequent polymerase chain reaction (PCR) tests and quickly identifying new cases [16–18]. Others have attempted lockdown areas to contain the spread of the virus. However, the effectiveness of these measures varies significantly, with many countries experiencing delayed epidemic detection and inefficient disease control. The effectiveness of frequent PCR testing and strict lockdown has been demonstrated in China, Australia, and New Zealand [12,19]. However, with the emergence of a more infectious Omicron variant, strict lockdowns have become inadequate. At that time, residents began to protest against the lockdowns, and the governments terminated the lockdown policy without any consideration for a soft landing [20–22]. Many other countries, such as Japan, have not conducted sufficient PCR tests, making it difficult to understand the trends in infection [12]. In some countries, such as Sweden, lockdowns were not implemented, and the public was left to make their own decisions [23]. In these countries, the epidemic progressed despite residents' self-help efforts. Thus, comparing the differences in the effects of these measures is crucial. The results of a 2022 study on the impact of various measures in European countries with the highest patient numbers are summarised as follows [24]. This is compared with 2021; during this period, the more virulent Omicron variant was prevalent, worsening the situation. Social responses such as cancelling gatherings and shutting down transport companies proved effective and were inversely correlated with infection numbers. The effectiveness of these measures has been recognised since the epidemic's onset [25]. Enhanced surveillance is commonplace in all regions where it has shown efficacy. Furthermore, a German study indicated that combined general behavioural changes were the most effective non-pharmaceutical intervention [26].

The views about the effectiveness of the vaccine vary; specifically, some studies suggest that the vaccine reduces both deaths and infections by 50–80% [27,28], while others suggest that there is no significant difference [29]. Nevertheless, there is a consensus that the vaccine is less effective against the Omicron variant. Supporting this, sequence analysis suggests that the Omicron variant has escaped immunity from a previous vaccine [30]. Both the numbers of vaccinations and boosters were strongly and positively correlated with the number of infections [24]. These results suggest that the vaccine is ineffective, at least against the omicron variant, and that vaccination may increase the number of infections. Furthermore, there has been concern that repeated vaccinations might lead to a decrease in immunity [31,32]; this has been confirmed to manifest as an increase in immunoglobulin (Ig) G4 [33–35].

Technical difficulties in identifying trends in the patient population, which can increase or decrease rapidly, may account for the failure to detect epidemics. Thus, adaptation to the susceptible–infected–recovered (SIR) model proves challenging, and it targeted a relatively limited number of individuals; the period of logarithmic increase was brief, rendering the estimation of $R_0$ [36] less pertinent, as viruses not initially human gradually acclimatised to humans. Those that become more infectious repeatedly penetrate defence systems and cause epidemics [37,38]. When switching hosts, the mutation rate is particularly rapid, and infectivity becomes stronger [39]. Therefore, the most recent Omicron variant, which has persisted for more than one year, is highly infectious [40]. Moreover, the basic reproduction number $R_0$, which the SIR model originally emphasised, has a lognormal distribution, is mathematically unstable, and requires complicated calculations for a precise estimation [36]. This makes it difficult to use as an indicator of trend. Therefore, it is more realistic to work with a logarithmic growth rate, $K$, which has a base value of two. This number represents the rate of change in the number of patients $N(t)$ at $t$ and is expressed as $N(t) = N_0 2^{Kt}$. More information has been published in previous studies [36] and can be found in Appendix A. This is easy to calculate and suitable as an indicator of the epidemic phase. The changes in $K$ are shown in the following data.

This study provided a comprehensive analysis of the effectiveness of various government policies in response to the COVID-19 pandemic. By identifying effective strategies for

containing the epidemic, this study can help guide policymakers in their decision making for containing COVID-19 and preparing for future pandemics.

## 2. Materials and Methods

### 2.1. Data

Data on the number of confirmed cases, deaths, and vaccinations were collected and published by the WHO [1]. However, for Japan, data published by the Ministry of Health, Labour and Welfare of Japan were used [41]. Furthermore, the mortality rate per 100,000 people [42] was used to determine the impact of the vaccines on mortality.

### 2.2. Calculations

The calculation principles are described in a previous study [36]. $K$ is calculated from the 7-day difference, $K = \log_2(\text{Cases}/\text{Cases}_{-7\text{day}})/7$, and the mortality rate is calculated from the ratio of the 18-day difference, mortality $= \text{Death}/\text{Cases}_{-18\text{day}}$. Each dataset was taken as a 9-day moving average for ease of viewing. Analyses were performed using the free statistical package R ver 4.3.0 [43]. The code and sample data used are available in the Supplementary Materials and can be used to reproduce all figures and results.

To compare the mortality rates with and without vaccination, the mortality ratios were calculated as follows: $r = \text{ratio}_{\text{without}}/\text{ratio}_{\text{with}}$. As the data were collected by age, the ratios were calculated for each age group and displayed as a boxplot. Because mortality rates vary on a logarithmic scale, this ratio is represented on the logarithmic axis. When one of the ratios was zero, $r$ became incalculable, or zero, because the amount of data was insufficient; these were omitted.

## 3. Results

### 3.1. Reliability of Observations

Prior to a detailed investigation, a concern must be acknowledged: data published by the WHO might deviate from reality, and the extent of this deviation varies greatly from country to country. This analysis necessitates accurate measurements [10]. For example, Japan has a relatively low number of PCR tests [12], which leads to a much lower estimate of the number of positive cases. The actual number of deaths estimated from excess deaths was approximately six times higher than that in published figures [6], and the same may be true for confirmed cases. Concerning other countries, a considerable proportion of the population in the USA, the United Kingdom, and Sweden may now be immune. However, reports from these countries comprise only half of the records for Iceland (Table S1). Moreover, there are concerns regarding improper testing, especially when the reported number of infections is low [44,45]. Surveillance in these countries should consider the results of near-random sampling rather than the entire sample size. Particularly, India and South Africa had even smaller proportions reported, indicating that possibly less than 10% of the population was investigated. Surveillance in these countries likely does not encompass a random sample but may be confined to the affluent. Should this be the case, the data cannot be equated with data from other countries. Conversely, Iceland is regarded as well surveyed [17,18]. These differences should be considered in the following studies.

### 3.2. Global Data as an Example

Refer to Figure 1A for a global illustration of how to interpret the figures. Large epidemics occurred in Alpha and Delta, but Omicron, which is thought to have finally acclimatised to humans [40], caused the largest epidemics with three waves in 2022–2023. The second wave affected many countries, including Oceania, and the third, more pronounced peak wave was primarily in China (Figure S1T,U). $K$ sharply increased prior to large epidemics. Sustained negative values of $K$ were necessary for several weeks to significantly reduce the number of infected individuals.

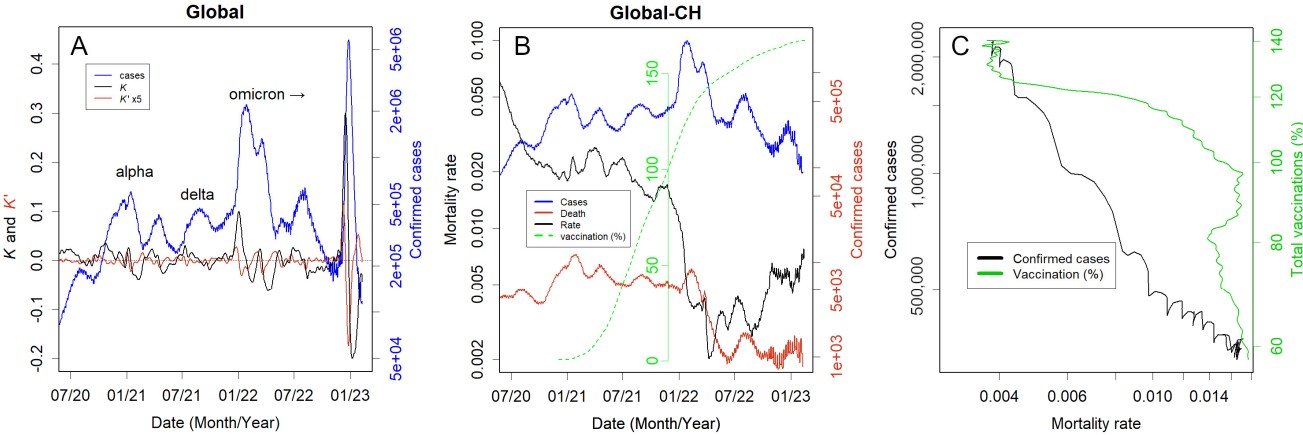

**Figure 1.** Summary of the global epidemic. (**A**) Confirmed cases of the epidemic (blue, right axis, logarithmic) and *K* (black, left axis). Confirmed cases are clearly reflected in weeks after *K* rises or falls. The third wave of Omicron, when *K* was at its highest, is almost exactly identical to that from Chinese data. Red is d*K*/d*t*, which rapidly shows changes in *K*. (**B**) Number of deaths (red, right axis) and mortality rate (black, left axis). Because data on the number of deaths are unavailable, China is excluded. Hence, the third wave of Omicron almost disappeared (blue, right axis). The green dashed line is the number of vaccinations (per 100 persons). (**C**) Change in mortality rates (x-axis, logarithmic) against the prevalence of the Omicron variant (black, left axis, logarithmic) and against change in vaccination rates (green, right axis, logarithmic).

Figure 1B summarises the mortality rate, namely the number of deaths relative to confirmed cases; this is calculated as the number of deaths divided by the number of people infected 18 days earlier (the average peak delay). The calculations here exclude China, which experienced the most significant impact. This rate changed on a logarithmic scale from approximately 20% at the pandemic's onset to 2% in 2021 as medical care adapted to the disease, and it decreased by an order of magnitude with the Omicron variant outbreak [46], but then increased again and was approximately 0.5% in 2023. The apparent partial overlap between the decrease in mortality and the vaccination coverage was coincidental. In strict terms, no effect was observed until vaccination coverage was close to 100%, and by the time this reached 120%, mortality had fallen (Figure 1C, green line). However, causality is unlikely to explain this unusual phenomenon. Nonetheless, the increase in Omicron-confirmed cases and the decrease in mortality correlated well (Figure 1C, black line).

### 3.3. Statistical Nature of the Data

The extent of infection can be determined, for example, by the ratio of infected persons to the total national population, and the severity and impact of the disease can be gauged by the number of deaths per infected person. Although these values could be estimated, they were challenging to ascertain. The number of infected persons per population was normally distributed, showing a pronounced skew towards the lower class (Figure 2A). The figure describes a QQ plot comparing the normal distribution with the ordered numbers. This is explicable if the aggregate of various measures contributes to the number of infections, in line with the central limit theorem. However, as of February 2023, even during the prevalence of the Omicron variant, approximately 100 countries reported near-zero numbers, indicating a skewed distribution. This may be because insufficient testing in these countries means that infected persons are not identified.

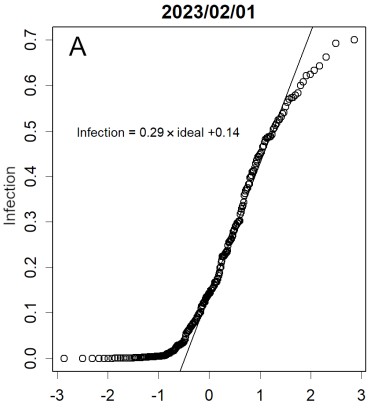
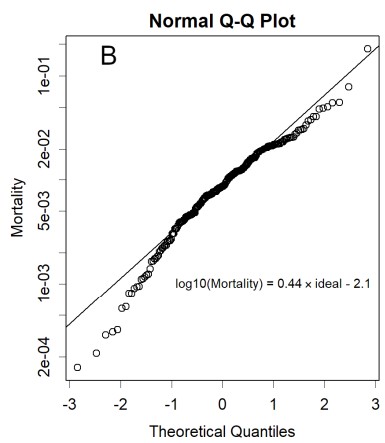
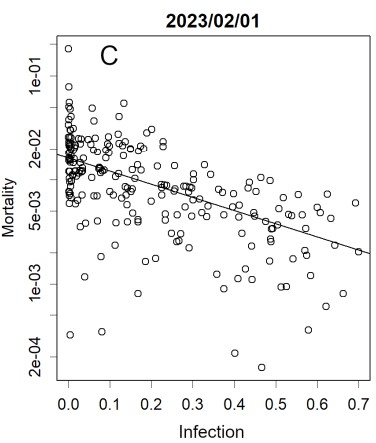

**Figure 2.** Ratio of infected individuals and mortality. (**A**). The ratio of infected individuals per population. Data for February 2023. This QQ plot compares the same quantiles of two data: the ideal values of the normal distribution are on the x-axis, and the sorted ratios are on the y-axis. Although there is a broad linear relationship, there is a large group at the bottom, mostly in the developing countries of Africa and Asia. These countries probably perform little testing. (**B**). Mortality: the number of deaths among positive COVID-19 patients; the average value of data from March 2022 to January 2023 is shown. A QQ plot shows the sorted mortality on the y-axis and the ideal value of a normal distribution on the x-axis, where the y-axis is expressed logarithmically. A linear relationship indicates a lognormal distribution. The straight-line curves downwards when the z-score is approximately −1. These are island countries with small populations, indicating an advantageous environment (Table S1). (**C**). Comparison of mortality and infection rates in a scattered plot. A weak and clear negative correlation was observed (Pearson's $r = -0.59$, *p*-value $< 2.2 \times 10^{-16}$).

The mortality rate per infected person was normally distributed (Figure 2B). This is understandable given the multiplicative effect of several factors: the line curves downwards from a z-score of −1, suggesting that there is something fundamentally different in these countries compared to others (as in the case of Figures S1C and S2C). Many of these countries are islands with small populations; however, some, such as Singapore and Iceland (Figures 2 and S2F,I), exhibit large populations.

Mortality rates increased when healthcare systems collapsed; however, there was a weak negative correlation between infection and mortality rates (Figure 2C). Many countries with extremely high mortality rates were not adequately tested. Perhaps in these countries, testing is conducted only in critical cases. As the burden of medical care increases, the mortality rate should rise; however, if adequate testing is not performed, an infection explosion may occur unnoticed, and medical care will be disrupted. Mortality and infection rates were z-scored according to their distributions (Table S1). This is an indicator that permits direct comparisons between countries and is a characteristic of each country. However, one cannot judge everything based on these alone, as the accuracy of the data is often questionable. The low mortality and high infection rates in some small island countries suggest that these countries were well screened and controlled. Nonetheless, these countries do not have large cities, and their borders are easy to control, which is not an option for other countries.

*3.4. Country Responses*

The responses of different countries to COVID-19 have varied widely. However, some countries have similar characteristics in terms of infection. The samples are shown here. Although statistical tests are often performed to demonstrate a higher degree of objectivity, only a few results are described as examples in the Appendix B. This is because when there is a marked difference under such a large n number, the test becomes somewhat redundant.

Before analysing the situation by country, I will first describe the relationship between *K* and the number of patients in Figure 3. Black represents *K*, and green represents the

moving average of *K* (the average from one month prior to the day). In Iceland (panel A), *K* was elevated from September 2021 to March 2022, leading to the largest peak in patient numbers. Subsequently, in April, *K* was successfully reduced to a minimum, resulting in suppressed patient numbers. *K* was elevated in May and June 2022, but by August 2022, *K* was lowered to nearly −0.1, further reducing the peak. Thereafter, *K* remained low, and patient numbers stayed low. Although I will not intensively discuss *K*′ here, it serves as a useful indicator of *K*'s trajectory. Even if *K* is high, it will eventually decrease if *K*′ is negative. Conversely, if *K*′ is not negative, *K* will increase or remain the same.

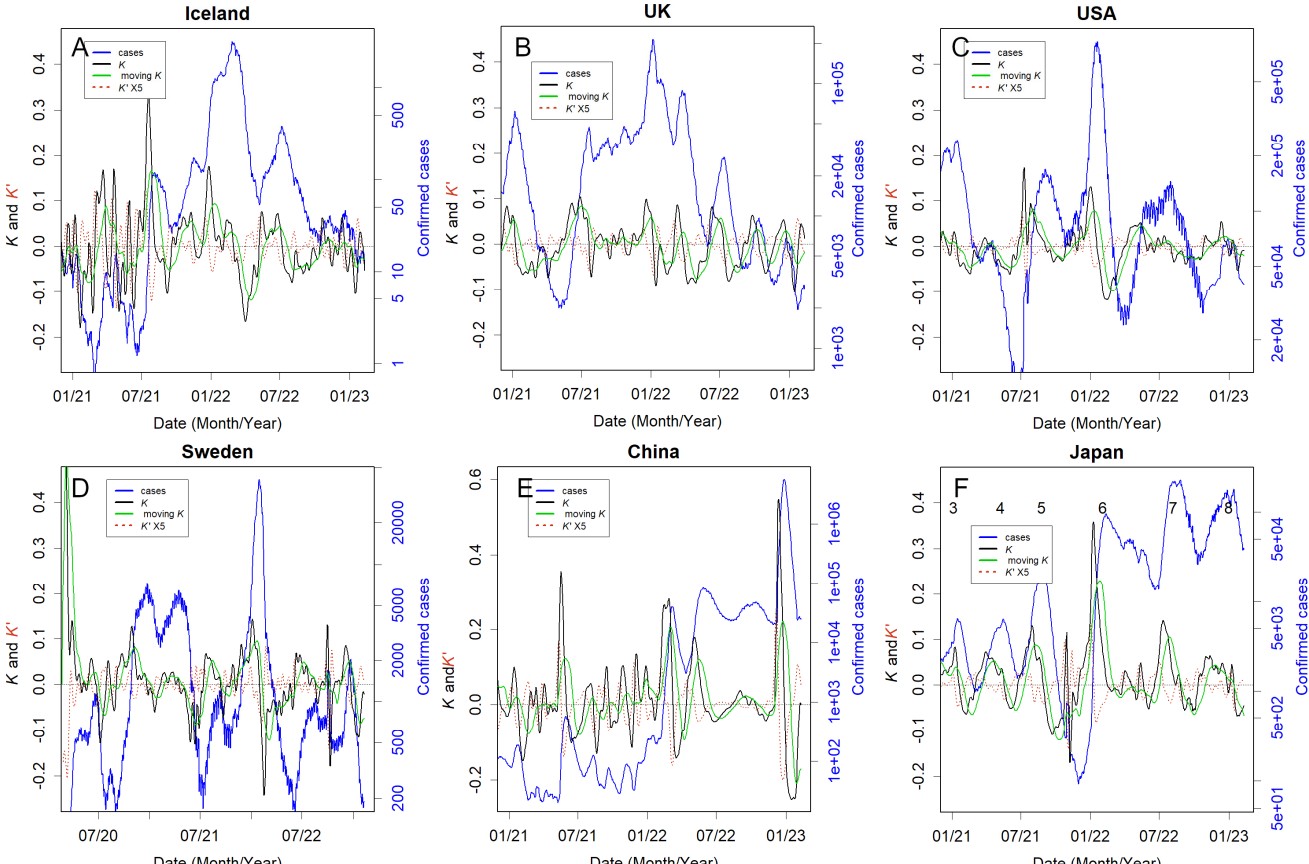

**Figure 3.** Confirmed cases for each country. See Figure S1 for data for other countries. Blue is the number of patients, black is *K*, green is the moving average of *K* (average from one month prior to the day), and red dotted line is *K*′. (**A**) Iceland: note the smaller level on the right axis; *K* became temporarily large but quickly converged, preventing the logarithmic increase phase from being long. (**B**) UK, (**C**) USA, and (**D**) Sweden: The Omicron epidemic of January 2022 was huge and might have infected many people who remained susceptible. Therefore, the second and third waves were smaller. (**E**) China: the large and rapid third wave of Omicron January 2023 appeared just after the lift of strict lockdowns. (**F**) Japan: There have been eight waves until now. The measurement of the eighth wave appeared to be somewhat inaccurate, perhaps because it did not clearly confirm the rise in *K*, and there could still be many susceptive people, and the three waves, 6, 7, and 8, were equally large.

In contrast, in the UK (Figure 3B), *K* remained positive for nearly one year from May 2021 and never became significantly negative. In the USA (Figure 3C), *K* remained positive for an extended period from June 2021 to mid-January 2022, triggering an explosion in January 2022. Once the outbreak subsided, *K* decreased to −0.1. However, in April and May 2022, *K* stayed positive, leading to an outbreak from April to July. In Sweden (Figure 3D), *K* did not decrease between July 2021 and February 2022, likely causing a peak in the February 2022 outbreak. Since then, both *K* and the number of infections have remained low. The Chinese lockdown failed to sufficiently control the rise in *K* from November 2021

to June 2022 (Figure 3E) due to the Omicron variant. This resulted in numerous patients being treated from February to November 2022. Immediately after lifting the lockdown in December, *K* increased sharply, with a significant peak observed in patients within that month. From July to August 2021, coinciding with the Tokyo Olympics, *K* remained high in Japan (Figure 3F). They managed to dramatically reduce *K* in November 2021 after the fifth wave of Delta variants ended, likely due to vaccine effects, leading to a decrease in patient numbers. However, with the emergence of the Omicron variant, *K* rose sharply in January 2022, causing a large number of infections. Since then, the inability to reduce *K* has continued to result in more infections between waves.

The mortality rate fluctuated continuously and logarithmically (Figure 4). In all countries, it temporarily decreased during major outbreaks of the earlier Omicron variant. However, this decrease was not sustained, and it subsequently increased again (Figure 1B). During large infection explosions, probably due to healthcare system breakdowns, it tends to rise.

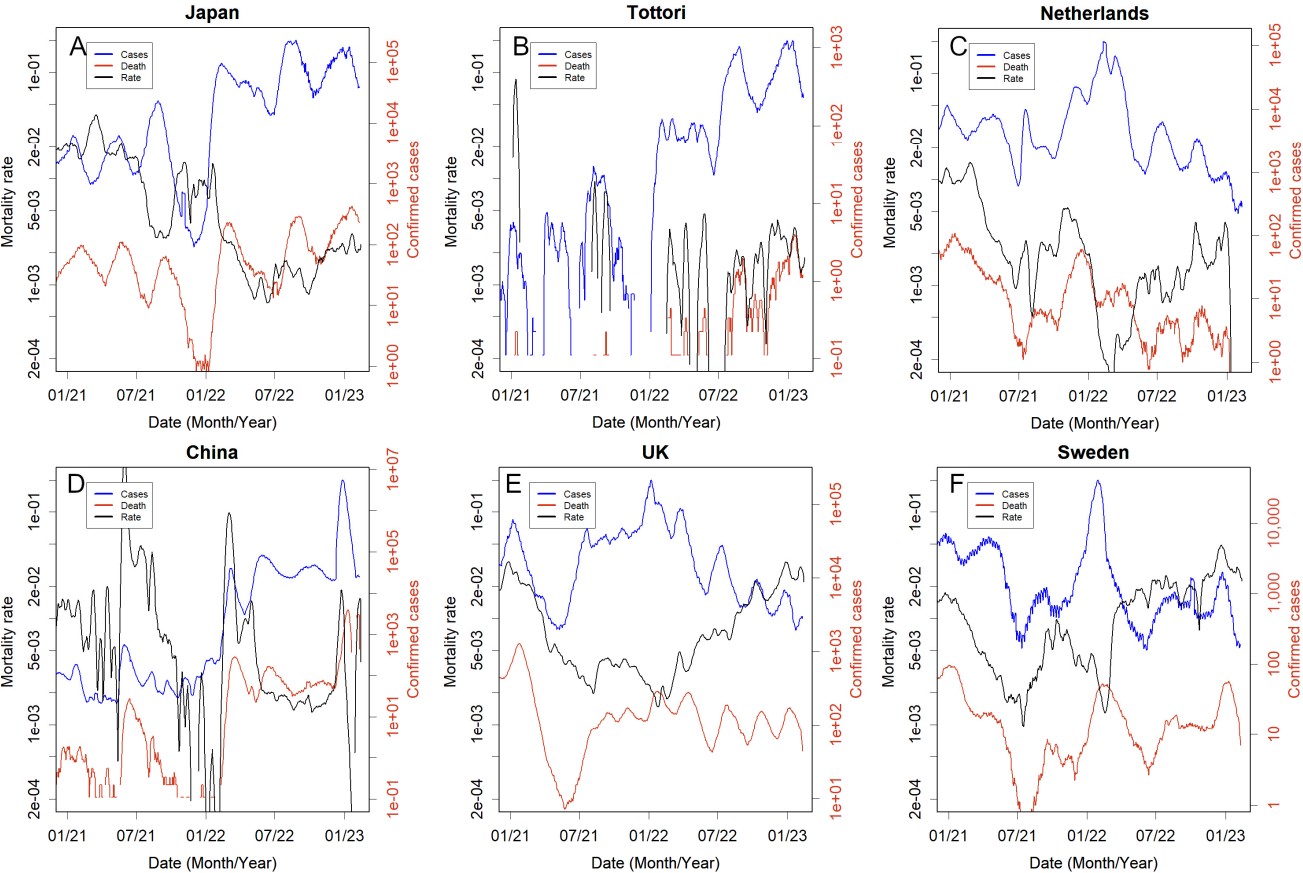

**Figure 4.** Number of deaths (red, right axis, logarithmic) and mortality rate (black, left axis, logarithmic). (**A**) Japan: when vaccination succeeded in reducing the number of confirmed cases, the mortality rates increased. (**B**) Tottori Prefecture (Japan) and (**C**) the Netherlands. The mortality rates are stable and low, remaining around 0.1%. (**D**) China: large epidemics have been observed since the lockdowns were lifted. Despite this, mortality rates have not increased, which is probably an error due to neglected measurements. (**E**) UK and (**F**) Sweden: after the Omicron outbreak, the number of infections decreased, but the mortality rates rose to 2%, which is four times the global average.

## 3.5. Countries That Took the Necessary Measures

The basic principle of infection control is preventing the emergence of new spreaders. As no vaccine is available to reduce new infections (Figure 1B) [46–48], this can only be achieved by isolating spreaders from susceptible populations [49]. This requires testing to determine who is infected and the people with whom they have come into contact, which,

in the case of COVID-19, requires constant PCR testing [45,50–52]. Therefore, facilities are required to treat patients in isolation. Iceland and Tottori prefectures in Japan are probably the best examples that have been able to implement both practices for a long time (Figures 3A and S1K). Even when $K$ increases in these countries and regions, it converges relatively quickly. This implies that a logarithmic increase requires less time; therefore, the number of infected people is less likely to increase. Taiwan is undoubtedly another country that has continued to adopt this ideal approach [53]. Unlike Iceland and Tottori, which have relatively small populations, Singapore and Taiwan have populations of over 20 million; therefore, any country should be able to learn from Singapore and Taiwan [16]. Data for Taiwan cannot be represented here because the WHO treats it as being included in China. Those of Singapore are included in the Supplementary Materials (Figures S1F and S2F). The Netherlands (Figure 4C), Israel, Denmark, and Switzerland (Figure S2H, S2S, and S2P, respectively) were included in this group (Table S1). In particular, the low mortality rate in the Netherlands could be largely attributed to the prevalence of a less lethal variant of mink origin by 2021 (Figure 4C, Table S1) [39]. Among the countries analysed, France and Germany performed well (Figure S2D,E). These countries succeeded in maintaining a low mortality rate, with z-scores of less than −0.6 (Table S1). The percentage of infected people was approximately 50%, with a z-score of approximately 1, which indicates the top 16% among countries (Figure 2A). The low mortality and high infection rates provide evidence that inspections could be conducted well, and the data would be relatively reliable.

### 3.6. Countries That Could Not Take the Necessary Measures or Made No Attempt to Do So

Leaders in the UK and USA failed to apply scientific knowledge [54,55]. This led to several continuous outbreaks (Figure 3B,C). Sweden became notorious and was criticised for its inaction at the outset [23]. However, after the Delta variant outbreak, the infection subsided in Sweden, probably because the public took self-defence measures such as social distancing (Figure 3D) [23]. Nevertheless, the Omicron variant, which is highly infectious, can circumvent these self-defence mechanisms, thus causing significant epidemics. The initial wave in these countries was substantial, and a considerable number of individuals likely acquired immunity; consequently, the subsequent waves were less severe, reducing the likelihood of another major epidemic (Figure S3C). Russia, with consistently high mortality rates throughout the pandemic (Figure S2M), probably belongs to this category.

Viral genome sequencing, incidentally, is unrelated to the identification and isolation of patients and, therefore, does not aid in preventing epidemics. Regardless of the variants that emerge, the tasks necessary to identify and isolate patients remain consistent. Indeed, sequence banks are inundated with almost identical data from the USA and UK. Therefore, extensive reliance on sequencing is unjustified. Instead, examining samples from countries without sequencing efforts is more sensible. It is certain that, sequencing is instrumental in identifying new variants. The Omicron variant, for instance, was discovered late because it emerged in a country lacking such efforts [40]. Ultimately, new variants from these countries are likely to cross borders.

### 3.7. People Defended Themselves without Recourse to the Country

The same is true in Japan, where the leadership took only unscientific and ad hoc measures [12]. People stopped cooperating with the quarantine because infected people were left alone in hotels without proper medical assistance [56,57]. For this reason, people have taken protective measures, and it has become customary for individuals to wear masks in public. Consequently, it is presumed that a substantial portion of the Japanese population has not been exposed to this disease; indeed, antibody test results using blood samples from blood donations indicate only about 40% positivity [58]. The Omicron outbreak is characterised by three distinct waves: a large number of people remain susceptible, and the quarantine system is not stringent, potentially leading to outbreaks from neighbouring countries (Figure 3E,F). A similar scenario may have occurred in Chile, where reliance on vaccines was high, yet people continued self-protection measures. Chile managed to

reduce *K* between infections, likely a result of these self-protection efforts (Figure S1B). Many individuals are not immune (Table S1).

*3.8. Countries That Relied Too Heavily on Lockdowns*

Several countries, such as China, Australia, and New Zealand, have implemented strict lockdowns but could not maintain their policies. After the lockdown was lifted, *K* immediately increased, and many infected individuals were detected (Figures 3E and S1A,L). The most dramatic rise was in China, where *K* reached 0.56, a tremendous rate with a doubling time of only 1.8 days, which is the highest ever recorded. However, the exact number was unknown as they had already stopped counting strictly (Figure 3E). Media reports suggest that approximately 90% of the population has been infected [59–61]. Moreover, there was a significant dissociation from the numbers confirmed by the WHO statistical data (Table S2). In addition to lifting lockdown policies, these countries have abandoned checking the exact daily status of infection [1].

*3.9. Countries Where Nothing Could Be Done*

The results in Figure 2A show that a considerable number of countries have reported few or no cases of infection. Many of these are developing countries in Africa and Asia, with high mortality rates (Table S1). Some might be an accurate representation of the situation in fortunate island nations. However, with migration, numerous countries are unlikely to avoid the spread of the infection. Many nations reporting few cases likely have not implemented measures. The total number of deaths attributable to COVID-19 remains unknown in such countries.

*3.10. What Difference Do the Policies Make?*

Differences in the measures altered the death toll. However, when examining these data, one needs to consider how well they reflect reality. For example, the data from India and China may have been inaccurate (Tables S1 and S2). The actual number for Japan is thought to be at least six times the reported number [6], which would be 0.3–0.4% deaths in the population; the eighth wave (the third wave of Omicron) was even more poorly measured [62,63], and the figure could be even higher [64]. The reported value may be higher in the United States also [6,44,45]. In contrast, Iceland's 0.064 and Tottori's 0.044% were probably more reliable.

The mortality rates were not constant but varied on a logarithmic scale (Figures 1B, 4, and S2), corresponding to the lognormal distribution of this rate (Figure 2B). This variation was attributed to the prevailing variant characteristics at the time and the changing burden on medical personnel, likely contributing to the tendency of mortality to increase as the epidemic expanded. As depicted in Figures 1B,C, 4 and S2, the sole epidemic during which the mortality rate declined was the first Omicron epidemic. Conversely, in other epidemics, the mortality rate escalated; for instance, the second and third waves of Omicron infection heightened mortality in Japan (Figure 4A). In contrast, Tottori's mortality rate remained stable, less than 0.1% (Figure 4B, Table S2), representing a stark contrast even within the same country (Appendix B) [65].

Since quitting the zero-COVID policy, Australia and New Zealand have abandoned the proper counting of patients or deaths [1,66]. While the number of patients was large, the mortality rates were low (Figure S2A,L, Table S1); however, many deaths have been reported in the media [67–69]. In China, when the Omicron variant slipped through the strict lockdown and became an epidemic, the mortality rate temporarily increased to 10%; this reduced incredibly and remained at a favourable 0.2% (Figure 4D). However, following the termination of the zero-COVID policy, when the latest death count was reported, it had risen to approximately 5%, a rate commonly seen in medical collapses [36].

### 3.11. Vaccine Effectiveness on Mortality

The vaccines were no longer effective in preventing infection (Figure 1B). This is evident, for example, when compared with the effects from October to December 2021 in Japan (Figure 3F). Then, it is imperative to assess whether the vaccines continue to exert a mitigating effect on mortality rates. Few published data are available to compare the mortality rates with and without vaccines worldwide: Japan [70], the USA, Switzerland, and Chile [42]. Among these countries, Chile predominantly uses multivalent vaccines (Sinovac Biotech Ltd., Beijing, China), whereas other countries use mRNA vaccines. An independent website compiled the data [42] that can be referred to, but neither country was active in publishing raw data; hence, statistical tests could not be performed. Figures 5 and S4 show the number of deaths as the ratio of vaccinated/unvaccinated individuals in these countries. This compares the mortality ratios between those who received the vaccine and those who did not. If the vaccine is effective, the number will be greater than one. The axes are plotted logarithmically. The mortality rates differed significantly during the delta epidemic in the United States, Switzerland, and Chile (Figures 5 and S4). An exception was Japan, where data were published only once during the delta outbreak (Figure S4A) and showed no significant difference ($p$-value = 0.423, paired $t$-test on logarithms of the ratios for age classes). With Omicron, smaller differences were observed in the USA (Figure S4B,C); however, this difference was no longer evident in Switzerland (Figure S4D,E) and disappeared in Chile (Figures 5 and S4F–H). Especially in the US, one should keep in mind that this is a comparison between vaccine recipients and anti-vaccine individuals; hence, a confounding issue may occur.

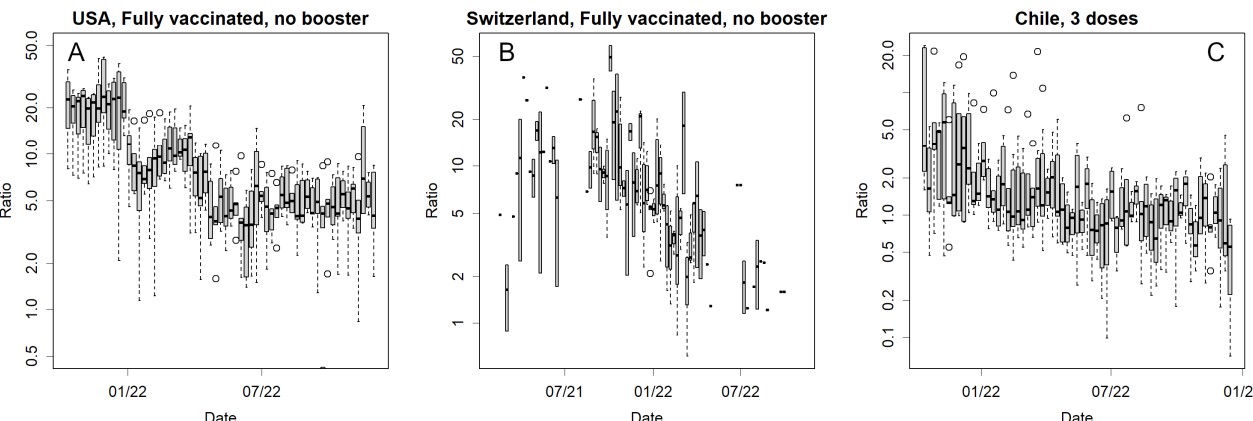

**Figure 5.** Comparison of mortality rates with and without vaccination. (**A**) The USA and (**B**) Switzerland: no booster vaccinations. (**C**) Chile: comparison between (zero or one) and three doses. The higher the ratio, the more effective the vaccine is in reducing mortality. For other data, see Figure S4. The box-and-whisker diagram is written by default in R. That is, the horizontal line at the centre of the box is the median, and the top and bottom of the box are the quartiles respectively. The dotted line is drawn when the data is at 1.5 times the length of the box. The dots indicate outliers greater/less than that.

Another way to estimate this is to examine how mortality rates change before and after vaccination. Globally, the mortality rate decreased following the widespread availability of the vaccine (Figure 1B). This decline, however, could potentially be attributed to the predominance of the Omicron variant (Figure 1C). Contrarily, as delineated in Figure 4, an increase in the mortality rate is observed across all countries. Specifically, in Japan, the mortality rate remained elevated even in December 2021, a period marked by the effective implementation of the vaccine and a reduction in infection rates (Figure 4A), and vaccines are not anticipated to significantly reduce mortality rates.

*3.12. Convergence of the Epidemics*

The development of immunity after an infection can prevent reinfection. This is why small epidemics repeat (Figures 1 and 3) and why the situation can be explained using the SIR model [36]. The epidemic converged, leaving a proportion of the population uninfected (Appendix A). This was because the population no longer represented a sufficient proportion of susceptible individuals. Similar to influenza, a sufficiently mutated variant can infect even immune people [71]. Indeed, some people are infected multiple times; however, they are less infectious, with only a smaller basic reproduction number, $R_0$. Every few years, an influenza epidemic occurs, but a novel virus such as pdm09 was able to cause a pandemic, and the scale of its influence was extraordinary [71]. Perhaps due to its nature, COVID-19 is less likely to cause major epidemics in certain countries (Figure S3). Moreover, the risk of serious illness due to reinfection is low [72]. Therefore, the number of deaths can be mitigated by preventing an outbreak of infection and alleviating the burden on medical care until a majority of the population acquires immunity. Consequently, Iceland has ceased employing stringent quarantine measures, and infection rates remain low. This suggests that the country has reached a state of convergence. Nonetheless, determining the conclusion of an epidemic based solely on patient count data describes challenges. For example, as illustrated in India (Figure S3B), in March 2023, there was a resurgence indicated by an increase in *K*, and the number of confirmed cases is currently escalating, as shown in Figure S1G. The cause of this trend, whether it is due to a new strain evading existing immunity or the presence of a substantial number of susceptible individuals, remains ambiguous based on these data alone, as indicated in Table S1.

**4. Discussion**

Strict policies, such as the stern lockdowns implemented in China, Australia, and New Zealand, proved to be unsuitable for controlling the pandemic, which is a battle expected to be protracted (Figures 4E and S2A,L). Moreover, a soft landing strategy is necessary to gradually phase out these stringent measures [20]. The isolation of infected people can be achieved via lockdowns; however, this places a heavy burden on the population, and it should be noted that Iceland did not opt for this [17]. Naturally, many citizens opposed the lockdown, which put pressure on politicians [73]. Eventually, the leaders of these countries, who had well-controlled epidemics via strict lockdowns, withdrew their zero-COVID policies [19,21,22]. However, this led to an epidemic (Figures 4E and S2A,L); it is likely that people accustomed to the lockdown had no means of self-defence. A few countries have stopped tracking the number of affected patients [1]. In contrast, Sweden continues this practice, enabling inspection of the mortality rate (Figure 3D). However, other disheartened governments may forsake necessary actions. This represents an additional drawback of lockdowns. The perceived success in controlling infection could result in overlooking the critical establishment of patient detection methods. This renders early treatment unfeasible and impairs infection control post-lockdown. Pertaining to other countries, lockdowns in Europe and the US have not been successful [15]; similarly, all measures in Japan failed (Figures 3, 4, S1 and S2) [12]. Another reason why countries that took more moderate measures did not succeed is probably due to the fact that they started too late. The peak of *K*, which showed an exponential increase, occurred several weeks earlier than the peak of infected individuals (Figures 3 and S1, Appendix A). By the time politicians began to take action, the infection was already on the verge of convergence, regardless of whether they had taken action. For this reason, observations should be made using an easily calculable indicator such as *K*.

mRNA vaccines, an entirely new technology [74], effectively prevented older variants (Figure 3F) [12,75]. However, it was ineffective against the newer Omicron variant (Figure 1B), which has undergone the selection pressure of vaccines and is in vogue. Severe acute respiratory syndrome coronavirus 2 (SARS-CoV-2) underwent significant mutations during the pandemic [30]. Therefore, the mRNA vaccine, which initially showed great promise, is no longer effective in containing the pandemic and is unlikely to effectively

reduce disease severity. Vaccine designs cannot keep pace with this mutation. Altering the mRNA sequence was required [76]. In a molecular biology sense, it was supposed to be a simple operation [74], but it took much longer than expected, and the new antigens were not effective. This task likely describes challenging problems known only to the individual concerned. The difficulty in continuously altering vaccine designs to accommodate viral changes. Consequently, mRNA vaccines may only be viable for a limited duration, necessitating the formulation of varied policies.

mRNA vaccines are expected to reduce the severity of illness and mortality; however, this remains largely unrealised (Figures 1 and 4) [46,75,77,78]. The reduction in mortality rates was probably due to the low lethality of the early omicron variant [46] and not due to the vaccines (Figure 1C). In some countries, mRNA vaccines up to the Delta variant might have been able to reduce mortality (Figure 5). However, this effect was much weaker for the Omicron variant, and I wanted to obtain additional data to confirm this finding. Especially in the USA, the situation is somewhat unique; there is a difference in access to healthcare between those who received free and easily available vaccines and those who did not. The latter must include many anti-vaccine people [79] who do not trust modern healthcare systems. Delayed medical intervention for infections with highly virulent delta variants would increase mortality. This readily presumed difference confounds the epidemiological estimates. Therefore, these results [80] should not be accepted as they are presented. Indeed, at least on a global scale, the delta variant was in epidemic proportions when vaccination coverage reached 100%, but mortality was only marginally reduced at this stage (Figures 1C and S2U). In Japan, the mortality rate persisted at a high level during the delta variant epidemic, a period when the vaccine's preventative effect was most pronounced (Figure 5A). Those infected during this phase were predominantly unvaccinated individuals. Regardless of whether there was a decrease in mortality rates at that time, this phenomenon might have instilled in the medical community a misconception that the unvaccinated are at a higher risk of developing serious illnesses.

Reducing the number of anti-vax [81] people is a task for governments. Public trust in the government diminishes each time counterproductive measures are implemented, potentially leading to escalating distrust in healthcare and a probable rise in the number of anti-vaccination individuals. This scenario was previously witnessed in Japan. A panel of experts tasked with advising the government (although their recommendations are often overlooked) advocated for avoiding densely populated areas and wearing masks; nonetheless, anti-vaccination groups resist these suggestions. They assert their autonomy in decision making, disregarding expert advice on lifestyle choices. Regrettably, if such individuals contract the virus, they spread it and risk fatal outcomes if intervention occurs too late. To prevent an increase in such attitudes, it is crucial to engage in direct information dissemination and education, ensuring they are not overwhelmed. Legal backing is essential for enforcing mandatory quarantine of infected persons. However, isolating patients without proper medical care is unfeasible. Furthermore, persisting in endorsing vaccines of questionable efficacy can erode public confidence.

Paradoxically, due to the anti-vaccine sentiment surrounding vaccines, their efficacy might be unjustly overestimated. Initially, epidemiological comparisons between unvaccinated and vaccinated individuals pose challenges owing to confounding factors stemming from disparities in healthcare access. Alternatively, such comparisons could yield statistically significant differences. Nonetheless, it is pertinent to recognise that this might result from the amplification effect of a large sample size (n); being statistically significant and exhibiting a large effect size are distinct concepts. Had there been a substantial effect, a decrease in mortality rates would have been evident. Yet, this was not the case (Figures 1B,C and 4).

It is unnatural and questionable why governments have not been proactive in publishing data on mortality and vaccination despite their importance to public health. If vaccines actually reduced mortality, the greatest publicity for vaccination would have come from these data. In Japan, despite aggressive promotion by the government [82], there is now a

surplus of discarded mRNA vaccines [83]. Should the impact of the decrease in mortality already have dissipated, this would pose a challenge for the government, which has been called upon to account for the substantial unexplained budgets [82–85]. Vaccines have lost their efficacy in preventing epidemics (Figure 1B). If vaccines do not reduce mortality then there are no benefits but only risks [86–88] in current vaccinations.

It is believed that the efficacy of the vaccine wanes over time but can be restored by boosters [76]. This was true, at least in terms of the in vitro [30]. However, this simple and naïve view was regarded as apprehensive [31,32], and it is unclear whether in vitro titres are linked to the actual immune function. Furthermore, it was finally shown that repeated doses of mRNA vaccine can lead to a class change in antibodies to IgG4 [33–35]. This leads to a progressive decline in immunity. Therefore, an overreliance on mRNA vaccines for protection is essentially impractical. The vaccine failed to prevent the infection (Figure 1A). Furthermore, they did not diminish mortality rates (Figures 1B, 4, 5 and S3). Vaccines that fail to prevent infection may not confer immunity; which mechanisms operate to enable vaccines to prevent severe diseases?

As can be observed that the Omicron variant led to a significantly higher number of cases compared to previous variants (Figure 1A), it is important to note that variants can mutate and enhance their infectivity, even during prolonged pandemics. This diminished the efficacy of initial measures such as social distancing, mask-wearing, vaccination, and lockdowns. The initial SARS-CoV-2 strain identified in this epidemic was not a human-specific virus; instead, it might have been a bat virus that, having been sustained in Vero cells, became infectious to primates. During human-to-human infections, the virus rapidly mutates to acclimate to humans in several directions [37,38], one of which is Omicron [40], the variant with the highest infectivity. This variant led to a reduction in the mortality rate; however, this was coincidental, as the virus was initially asymptomatic in many individuals. Consequently, weakening the virus does not undergo selective pressure. We should not invariably rely on such fortuitous outcomes. For instance, if the subsequent pandemic were to be influenza H5N1, it might manifest as a mutant capable of infecting mammals, transmitting via an intermediate host such as a pig, and subsequently infecting humans [89]. In this case, the virus changes rapidly in humans [39], altering its epitope and increasing its infectivity. However, if we can converge in the initial stage, we can avoid a pandemic before it occurs. This is the only stage at which a lockdown should be implemented. Therefore, a system that can quickly identify and alert the public to new infectious diseases is required.

The epidemic has subsided in some countries (Figures 3 and S3). SARS-CoV2 has several conserved open reading frames (ORF) [37,40], which is a major difference from influenza, where all ORFs mutate at equal rates [71]. Presumably, SARS-CoV2 cannot repeat reinfection, such as influenza, for decades. As people become less immune, multivalent vaccines, especially those that can be used in developing countries, are required [39,40].

Data on daily changes in confirmed cases are important for fully ending the epidemic. As the number of infected individuals declined, several countries stopped reporting daily data [1]. For instance, in the USA, data are reported solely on a weekly basis, whereas in Australia and New Zealand, the frequency has become increasingly irregular. This poses a significant public health challenge, as accurate estimation of $K$ is unattainable without consistent data. This situation leads to a rapid escalation and to an unobserved reduction, which is essential to sustain a decreasing trend. Per the SIR model, the patient count diminishes exponentially [36]. However, this phenomenon was not observed in these countries. Although the numbers are declining, small epidemics still recur, resulting in patient numbers ranging from tens to several thousands (Figures 3B,C and S1A,L). The exponential decrease is initially rapid but progressively slows down. Achieving complete convergence is challenging if the government cannot sustain its motivation to keep the $K$ low. While the development of effective vaccines and certain drugs is desirable, the government should not rely solely on them but prioritise immediate necessary actions. The

only way to resolve this issue is via the continued identification and isolation of patients, a responsibility that should lie with the government.

The inability to contain the causes of problems stems from the fact that the remaining patients comprised a larger proportion of vulnerable individuals. These may include persons with underlying diseases who have diligently avoided infection and those who, failing to acquire immunity, become infected repeatedly. This scenario was likely in individuals who had received multiple vaccinations [33–35]. Therefore, these patients were more likely to be critically ill [90]. This disease can cause systemic symptoms [91] and long-lasting sequelae [2–5,92] in over 10% of individuals, and this rate is expected to worsen in the future as the number of susceptible individuals rises. The UK, Sweden, and Denmark may not have enough patients to cause a medical collapse now; however, their mortality rates have remained high, at approximately 2% (Figures 4E,F and S2S). Even in the USA, where there may be a wider choice of medical care, the mortality rate is 1% (Table S1), which is quite high compared to the global data (Figure 1B).

The only viable method to protect vulnerable individuals is to bring the COVID-19 epidemic to a complete halt. Furthermore, the emergence of new strains is likely if outbreaks persist in these countries. Should these mutate significantly, they might precipitate another pandemic akin to the pdm09 strain of influenza that has engulfed the world [71]. Once the outbreak appears to have subsided, as observed in India, a new variant may commence circulation anew (Figure S1G). The continuation of measures and the detection of infected persons is imperative. The absence of such measures has led to a lack of convergence in Japan, with the number of infected individuals remaining unchanged, even between epidemic peaks (Figure 3F).

Numerous countries have underestimated the scale of the COVID-19 pandemic, although to differing extents (Tables S1 and S2). This underestimation likely stems from delayed detections, indicative of a shortfall in testing and isolation strategies. While the survey approached random sampling, a key limitation was the unknown sample size. Consequently, the precise number of patients and fatalities remains uncertain. In most African nations, there is a lack of investigation or response to infection cases; thus, the reported case numbers from Africa are markedly low (Table S1) [1]. This can only be estimated using certain methods. One promising method is to estimate excess deaths [6,67–69]. However, this estimation is contested; there is no substantiated evidence to confirm that the deceased were COVID-19 patients. For instance, an individual succumbing to a stroke may have perished due to the lack of prompt medical care. Nonetheless, it is probable that during normal circumstances, such an individual would have been rescued, thereby categorising their demise as a consequence of the COVID-19 pandemic.

Furthermore, epidemics can emerge in countries where minimal cases have been reported due to the absence of preventative measures and the evolving nature of variants during the epidemic's course. Data collated by GISAID reveal a pronounced bias in the geographic origins of these data; for instance, most African nations, with the exception of South Africa, have not contributed sequences. Scant records exist for the early Omicron variants or their precursors; it is speculated that the Omicron variant originated within the African continent, yet it remains unsequenced there [40]. From a humanitarian perspective, and in order to prevent pandemics, effective infection control measures are essential in these countries. A need exists for an international organisation or collaborative effort focused on sequencing and patient detection. Specifically, the development of a vaccine that can be independently administered in these countries, such as an attenuated virus vaccine, is highly desirable. Importantly, this vaccine should not compromise immunity, even after repeated inoculations.

The consequences are significantly more severe if there is an absence of leadership that pays adequate attention to public health or, at the least, heeds expert opinion (Tables S1 and S2) [54,55,65]. This escalation is independent of the country's guiding principles. A mechanism should exist whereby the scientific community can provide effective advice to the government. However, this system is not operating effectively in Japan.

In January 2023, Japan reported over 10,000 deaths, a number anticipated to multiply substantially [6,64]. Nevertheless, the government has made a cabinet decision to exempt COVID-19 patients from quarantine and is campaigning for people to stop wearing masks [93–95]. In addition to limiting the number of PCR tests, the government abandoned counting cases, stating that it would announce the number of deaths after two months [96], thus hiding issues from the people. Additionally, they are still promoting mRNA vaccination [82]. How unscientific these policies are is beyond dispute [33–35,86–88,97–99]; these unscientific policies will definitely affect many patients, including vulnerable ones. Clearly, the government was reluctant to assume responsibility for public health. One of the essential roles of experts is to scrutinise policies and persuade voters to avoid electing unsuitable leaders. In a country with a democratic system for electing leaders, voters must consider these crisis responses: Will the candidates enforce forced hardship, remain ineffective, or ensure intelligent identification and quarantine of infected individuals? An expert can discern these differences. The prompt removal of unscientific politicians is a necessary step for effective public health management.

Although reports of confirmed cases are decreasing as many countries have stopped taking action [1], COVID-19 does not appear to have been completely terminated, and new variants continue to emerge [9,30]. Moreover, this will not be the last plague; similar pandemic diseases are likely to emerge in the future, and it remains to be determined whether it is more effective to respond with detection and quarantine as described herein or if alternative methods should be considered on a case-by-case basis. Consequently, objective and accurate measurements and analyses are essential [10].

## 5. Conclusions

Large epidemics must be avoided as they can lead to a shortage of medical resources, potentially causing further damage. *K* should be monitored daily to detect and control outbreaks. An increase in *K* was observed during the epidemic; however, reducing it rapidly prevented the pandemic from becoming more widespread. The most crucial step involves identifying and isolating patients. Despite numerous efforts, it will be a considerable time before a vaccine or specific drug is developed; thus, excessive reliance on these measures should be avoided. Government officials should take appropriate actions to complete these tasks without significantly burdening citizens. Scientists should provide suitable advice on the policies that need to be adhered to.

**Supplementary Materials:** The following supporting information can be downloaded at: https://www.mdpi.com/article/10.3390/covid4020012/s1, Figure S1: world's data of confirmed cases, Figure S2: world's data of death, Figure S3: some confirmed cases in an antilogarithm axis, Figure S4: comparison of mortality rates with and without vaccination, Table S1: Mortality rate and Infection rate. Table S2: Total vaccinations per 100 people at the presented date, total infected (%), and death (%) on 2023/02/10.

**Funding:** This research received no external funding.

**Institutional Review Board Statement:** Not applicable.

**Informed Consent Statement:** Not applicable.

**Data Availability Statement:** All the data are available from homepage of WHO [1]. The R-code used is available in the Supplementary Materials.

**Conflicts of Interest:** The authors declare no conflicts of interest.

## Appendix A  The Character of K

According to the SIR model, the number of patients underwent exponential growth, followed by a decrease in the number of susceptible people. Because the recovery of infected patients can be approximated as a first-order response, the decrease was also exponential. For example, we simulated the cases of $R_0$ = 1.3, tau = 2 (Figure A1A). Here, *K* changed from 0.13 to −0.09; if the epidemic subsides once, *K* exhibits constant values as

shown in the panel A, with a single transition. Exponential growth or decrease occurred when *K* was constant. By the time people notice, the exponential growth slows, *K* is zero when the number of infections is at its peak, and the exponential decrease starts when the number of infections drops considerably.

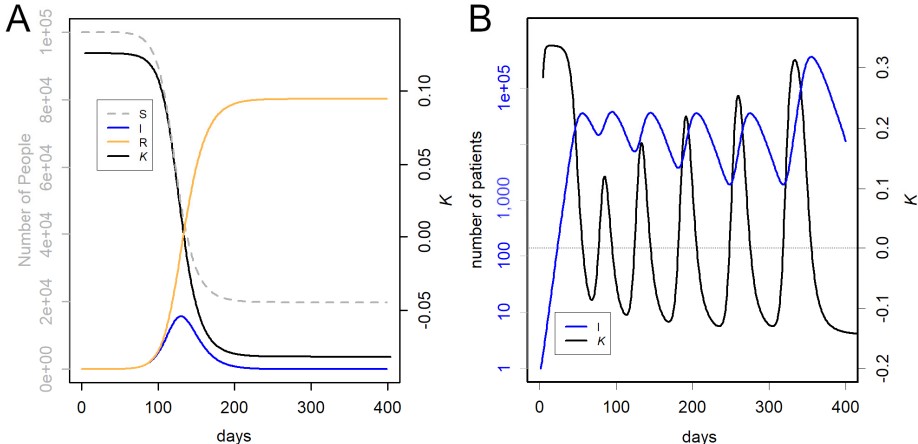

**Figure A1.** Simulation using SIR model. (**A**) Single epidemic. Gray, susceptible; blue, infected; orange, recovered. Black, *K*. (**B**) Repeated epidemics.

The exponential growth can be expressed using the following equation: At elapsed time *t*, the number of infected patients is $N(t) = N_0 2^{Kt}$.

Taking the logarithm of both sides of the equation

$$\mathrm{Log}_2(N(t)) = Kt + \log_2(N_0),$$

where *K* denotes the slope of the linear equation. In this study, this was estimated from the difference at one week. Because the exponential and logarithmic forms have a base of 2, $1/K$ represents the doubling time or half-life.

In the case of COVID-19, the infection has grown rapidly, the total number of patients is rather small, and only part of the population is infected. The next variant begins infecting the population before it completely converges. Therefore, both the logarithmic increases and decreases had short peaks, and *K* constantly changed (Figure A1B). For more details on the SIR model and simulations, refer to [36].

**Appendix B  Statistical Tests**

When n is large, as in the country-controlled data, the statistical tests have fewer positive implications. This is because the errors due to sampling are small and the data do not oscillate significantly. To illustrate this, a relatively small number of cases are presented. It looks at the proportion of deaths among infected individuals in Japan as a whole and in the Tottori Prefecture. Data up to the sixth wave in May 2022, when this small area had well-controlled infections, were compared with national data and tested using Fisher's Exact Test for Count Data. This method was used to test the two ratios. The number of deaths in Tottori was only 20, and the ratio was 0.42 times that of the entire country. The 95% confidence range of the ratio was 0.25–0.64, with a *p*-value of $8.7 \times 10^{-6}$ for the null hypothesis of no difference.

When n was larger, for example in mortality rates between Iceland and the USA, deaths in Iceland were 0.1 times that in the USA. The 95% confidence range for this ratio was 0.091–0.12, with a *p*-value of $<2.2 \times 10^{-16}$. This indicated that the *p*-value was too small to be calculated. It will be seen that when large differences are found from data of this large size, there is less need for testing.

In addition to being less necessary, there is another problem with statistical tests. This is a multiplicity of the tests. In such a large number of countries, the number of tests would be quite high if performed in a round-robin manner.

In some cases, the prior probability $Pr(H_0)$ may be close to one. This can be seen, for example, in quality control in factories because products are usually produced consistently. In this case, the test causes a false positive with a probability approximately equal to the threshold value. If such tests are repeated, the number of false positives inevitably increases. This is the multiplicity problem in the tests. In contrast, when we test the results of an experiment in which we expect some difference to occur, the prior probability $Pr(H_0)$ is close to zero. In this case, the probability of false positives is low. Therefore, multiplicity is not a major problem.

When comparing countries, it is ambiguous whether there should be a difference and $Pr(H_0)$ is unknown. Therefore, the suspicion that a multiplicity problem may arise when the test is repeated cannot be dispelled.

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
