# Peer review of "A Comparative Analysis of COVID-19 Response Measures and Their Impact on Mortality Rate"

_covid, doi:10.3390/covid4020012_

Round 1
Reviewer 1 Report (Previous Reviewer 3)
Comments and Suggestions for Authors
The issues of my interest have been resolved.
Author Response
I would like to appreciate your kind review. It has made the manuscript much easier to read and more accurate than the original. I thank the reviewer.
Reviewer 2 Report (New Reviewer)
Comments and Suggestions for Authors
Thanks for the opportunity to review the manuscript entitled ”Comparative analysis of COVID-19 response measures and their impact on mortality rate”.
I enjoyed reading this manuscript. The author makes a complex analysis of mortality rates in COVID-19 pandemic in different countries arround the world, by evaluating the capacity of detection and isolation of patients, and then by correlating between COVID-19 vaccines effects on morbidity and mortality due to this viral infection. The conclusion is interesting, that even of the designs of mRNA vaccines were changed and renewed, these could not keep up with the rate of virus mutations. Thus, it is most important to make efforts for detection of infection and the best measures to limit the spread should be the isolation of patients.
Author Response
I would like to appreciate your kind review. It has made the manuscript much easier to read and more accurate than the original. I thank the reviewer.
This manuscript is a resubmission of an earlier submission. The following is a list of the peer review reports and author responses from that submission.
Round 1
Reviewer 1 Report
Comments and Suggestions for Authors
-The introduction provides a clear context regarding the global impact of the COVID-19 pandemic and the importance of understanding the effectiveness of control measures. It effectively outlines the research problem and objectives. However, it lacks a concise review of existing literature on COVID-19 control measures, which could provide a better foundation for the study.
-The methodology section describes the approach, including the use of logarithmic growth rates "K" for assessing the impact of control measures on confirmed cases and mortality rates. While the methodology is clear, it could benefit from further elaboration on data sources, data preprocessing, and statistical techniques applied. Additionally, the paper should explicitly mention ethical considerations related to the use of pandemic data.
-The paper reports that countries effectively detecting and isolating patients had significantly lower mortality rates. This finding is consistent with expectations and aligns with public health principles. However, the paper lacks detailed statistical analysis and hypothesis testing to support these conclusions. Including confidence intervals and p-values would enhance the rigor of the analysis.
-The discussion provides valuable insights into the effectiveness of control measures, especially the importance of patient detection and isolation. However, the paper should discuss the limitations of the study, such as potential biases in reporting, variations in healthcare systems, and socio-economic factors affecting outcomes. Addressing these limitations would provide a more comprehensive view of the findings.
-The paper should be enriched with references to existing research papers on Data-Driven future of healthcare such as https://doi.org/10.58496/MJBD/2023/010. This would strengthen the argument and provide readers with additional resources for further exploration.
-To enhance the paper's impact, the author should discuss potential future directions for this research.
Reviewer 2 Report
Comments and Suggestions for Authors
The manuscript claims to study the impact of control measures on COVID mortality rates. The manuscript has very little actual analysis and is full of speculative and unproven claims. Some specific points:
1. Of the 78 citations, only 36 are peer-reviewed scientific publications. While most of the other citations are from news outlets with reasonable journalistic standards, some of these sources are somewhat shady.
2. The author calculates the growth rate (and its derivative) for a large number of countries, but then doesn't use it for any sort of analysis. Some statements in the manuscript about the growth rate contradict the graphs. For example, the author states "Even when K increased in these countries and regions, it con-
verged relatively quickly." Converged to what? From the graphs, K looks like it fluctuates more in these "converging" countries than in countries where it supposedly didn't converge.
3. Why would it be beneficial for K to converge anyway? If K converges to a positive value, that means the number of cases will continue to grow (and never decrease). Even if K converges to 0, that simply means that the case count is not changing --- if it stops changing when the case count is already high, it will remain at that high level meaning many people will get sick.
4. Lines 58-61: All indications are that the mRNA vaccines are still effective at preventing serious illness despite any mutation in the virus. The author is claiming otherwise, but provides no supporting references.
5. Lines 143-147: The author is making claims about the efficacy of Taiwan's response as compared to other countries. No data from Taiwan is presented in the manuscript or supplement.
6. How are groups of well-performing countries or poorly-performing countries being decided? What metric is being used?
7. There is no analysis attempting to correlate mortality to any lockdown measure, so it's not clear how the author is attempting to assess the claim in the manuscript's title.
8. Lines 151-153: "The fact that the percentage of infected people is approximately 50 % is evidence of the relatively low level of missed detection (Table S1)." I don't follow the logic here.
9. What is the reasoning behind the choices of which countries are included in Figs. 2 and 3? They don't all appear to be from the same "group", nor do they appear to be from different "groups". The countries represented in the two figures are different --- why did some countries make it into both figures, but others got changed?
10. Lines 167-169: "However, after the Delta variant outbreak, the infection subsided in Sweden, probably because the public took self-defence measures such as social distancing (Figure 2D)." This is just speculation. Where is the evidence that Swedes took more protective measures during this time?
11. Lines 175-182: This is a discussion of sequencing that appears to have nothing to do with the topic of the manuscript.
12. I'm not sure what the author is getting at with the analysis of vaccine efficacy. As long as r>1, the vaccine is protecting people from death. While r does decrease over time, it remains over 1 for both the US and Switzerland, so concluding that the mRNA vaccines are "ineffective" is incorrect.
13. I don't understand the what the author is trying to say in section 3.9. Are they suggesting that K is somehow predictive of the end of the epidemic? I don't see how that is possible.
14. Much of the discussion is speculation and not supported by any of the presented data. Some examples: Lines 315-358 discuss the effect of anti-vaxxers --- where is that studied in the data or analysis. Line 338 states that the infectivity of the virus increases --- where is the evidence for that? Lines 356-357 state that reinfection with SARS-CoV-2 is not possible, which is incorrect. There are several points where the author is suggesting government conspiracies that are keeping the truth from people.
Comments on the Quality of English LanguageThere are problems with sentence structure and occasional unscientific terminology.
Reviewer 3 Report
Comments and Suggestions for Authors
The authors have given a comparative analysis of CONID-19 response measures and their impact on specially mortality rate. It is very important.
However, some suggestions should be considered.
1) The logarithmic growth rate K was mentioned in this paper. What is the relationship between the value of K and the number of infected people?
2) Perhaps it is better to use the theory of optimal control to describe the effects of various response measures on the mortality rate, for example, the mortality rate as the control variable.
3) I recommend that your article should add more explanations on the charts ,table or some of the figures, for example (Figure 4,S4) in the line 254.
Comments on the Quality of English LanguageMinor editing of English language is required.